# Game of Stones: feasibility randomised controlled trial of how to engage men with obesity in text message and incentive interventions for weight loss

Stephan U Dombrowski ![ORCID],[1,2] Matthew McDonald,[3] Marjon van der Pol,[4] Mark Grindle,[5] Alison Avenell,[6] Paula Carroll,[7] Eileen Calveley,[3] Andrew Elders,[8] Nicola Glennie,[3] Cindy M Gray,[9] Fiona M Harris,[3] Adrian Hapca,[10] Claire Jones,[11] Frank Kee,[12] Michelle C McKinley,[12] Rebecca Skinner,[2] Martin Tod,[13] Pat Hoddinott ![ORCID] [3]

For numbered affiliations see end of article.

**Correspondence to**
Dr Stephan U Dombrowski;
stephan.dombrowski@unb.ca

## ABSTRACT

**Objectives** To examine the acceptability and feasibility of narrative text messages with or without financial incentives to support weight loss for men.

**Design** Individually randomised three-arm feasibility trial with 12 months' follow-up.

**Setting** Two sites in Scotland with high levels of disadvantage according to Scottish Index for Multiple Deprivation (SIMD).

**Participants** Men with obesity (n=105) recruited through community outreach and general practitioner registers.

**Interventions** Participants randomised to: (A) narrative text messages plus financial incentive for 12 months (short message service (SMS)+I), (B) narrative text messages for 12 months (SMS only), or (C) waiting list control.

**Outcomes** Acceptability and feasibility of recruitment, retention, intervention components and trial procedures assessed by analysing quantitative and qualitative data at 3, 6 and 12 months.

**Results** 105 men were recruited, 60% from more disadvantaged areas (SIMD quintiles 1 or 2). Retention at 12 months was 74%. Fewer SMS+I participants (64%) completed 12-month assessments compared with SMS only (79%) and control (83%). Narrative texts were acceptable to many men, but some reported negative reactions. No evidence emerged that level of disadvantage was related to acceptability of narrative texts. Eleven SMS+I participants (31%) successfully met or partially met weight loss targets. The cost of the incentive per participant was £81.94 (95% CI £34.59 to £129.30). Incentives were acceptable, but improving health was reported as the key motivator for weight loss. All groups lost weight (SMS+I: −2.51 kg (SD=4.94); SMS only: −1.29 kg (SD=5.03); control: −0.86 kg (SD=5.64) at 12 months).

**Conclusions** This three-arm weight management feasibility trial recruited and retained men from across the socioeconomic spectrum, with the majority from areas of disadvantage, was broadly acceptable to most participants and feasible to deliver.

**Trial registration number** NCT03040518.

### Strengths and limitations of this study

► This weight management study for men with obesity recruited men across the socioeconomic spectrum, with a specific focus on more disadvantaged areas, to a randomised controlled feasibility study examining SMS with or without endowment incentives.

► Acceptability and feasibility were established using a multilens perspective drawing on quantitative, qualitative and trial procedure data.

► Effectiveness of intervention components will need to be established in a full multicentre trial.

## BACKGROUND

Obesity is associated with an increased risk of serious health conditions such as type 2 diabetes, cardiovascular disease and some cancers.[1] In 2016, 26% of men in the UK were classified as obese, and men were less likely to be a healthy weight than women (31% of men compared with 35% of women).[2 3]

Despite the growing prevalence of obesity, men contribute a disproportionately low number of participants to evidence-based weight management programmes.[4] Most interventions are designed for mixed sex populations, but systematic review evidence suggests that, compared with women, men often benefit from different ways of providing interventions.[4]

### Text message interventions

The evidence for text message interventions supporting changes in lifestyle behaviours, including behaviour change for weight loss, is promising.[5–7] However, systematic reviews[4–7] report no text message delivered trials that target weight loss designed for men only. Text-based interventions can reach large

numbers of people, including men from disadvantaged backgrounds,[8] and mobile technologies such as standard mobile phones allow delivery of evidence-based strategies anywhere and anytime.

Narrative approaches to promoting behaviour change have been suggested as a tool for communication-based interventions.[9] Narrative SMS can be broadly defined as interactive life stories, which are based around a group of characters with whom recipients can identify.[10] Text message-based interventions using narratives have been used to engage hard to reach men in moderating their alcohol consumption and were found to be acceptable.[11–13] Narrative approaches to weight management in men may be a promising tool to support behaviour and weight change.

### Financial incentive interventions

Systematic review evidence of financial incentives for behaviour change highlights the potential for incentives to change behaviours in low-income adults[14] and help reduce health inequalities.[15] However, little evidence exists that focuses on using financial incentives to support weight management in individuals from disadvantaged backgrounds,[16 17] particularly for men.[4] A discrete choice experiment found that paying people (£10–£30/week varying by age and weight) to take part in diet and physical activity (PA) interventions is likely to improve uptake, adherence and maintenance of behaviour change.[18] Moreover, the evidence for financial incentives for weight loss is growing.[19–22] In particular, deposit contracts, where participants deposit their own money and are reimbursed only if they achieve the target weight loss, are effective while the incentives are in place.[4 23] Deposit contracts draw on loss aversion where people are more motivated to avoid losses than they are to achieve similarly sized gains.[24 25] However, deposit contracts may not be not equitable, as committing one's own money up front may not be possible, particularly for individuals with lower income.

Endowment incentive contracts may overcome limitations of deposit contracts. Endowment incentive contracts are financial incentives where a person is endowed with an (hypothetical) amount of money that they either 'secure' or 'lose' depending on achieving certain targets. A trial that framed financial incentives in this way found these incentives were effective for achieving PA goals in the short term.[26]

### Aim

The aim of the Game of Stones feasibility trial was to examine the acceptability and feasibility of a men-only weight management intervention consisting of narrative text messages, with and without an endowment incentive, compared with waiting list control. The objectives were to:

1. Assess the acceptability and willingness to be randomised to: (1) narrative text message and endowment incentives; (2) narrative text messages only; or (3) waiting list for text messages (control).
2. Assess the feasibility of recruiting from general practitioner (GP) practice obesity registers and community venues.
3. Determine the acceptability of intervention content, feasibility of delivery, fidelity and any unintended consequences.
4. Assess indicative effects on weight change and progression criteria for a full trial.

## METHODS

Consolidated Standards of Reporting Trials (CONSORT) guidance for reporting randomised pilot and feasibility studies was followed.[27 28]

### Trial design

A three-arm individually randomised parallel-group controlled feasibility trial was conducted with an integrated qualitative and quantitative mixed methods approach.[29] Informed by Medical Research Council (MRC) guidance on the evaluation of complex interventions, the study included an integrated mixed methods process evaluation.[30 31] Drawing on both qualitative and quantitative approaches allowed the team to explore participants' views (acceptability) of the intervention with participants as well as explore implementation processes such as recruitment, retention and barriers and facilitators to these.

Participants were randomised to receive narrative text messages and endowment incentives (SMS+I), narrative text messages only (SMS only) or a waiting list for text messages (control). The protocol to the full study is available online: https://www.journalslibrary.nihr.ac.uk/programmes/phr/1418509/#/

### Setting

Two health board areas in Scotland (sites A and B) with high levels of disadvantage were the setting for this study (see Participant recruitment section for additional details).

### Eligibility criteria

Participants met the following eligibility criteria: men over 18 years old; body mass index (BMI) $\geq 30$ kg/m$^2$ and/or a waist circumference of $\geq 40$ inches (102 cm); owned a mobile phone capable of receiving text messages; could understand English language text messages; not already taking part in a weight loss study; not planning or waiting to have bariatric surgery; not planning to move within the next 12 months; and considered by practice clinical staff as suitable for participation (GP practice recruitment only), for example, no severe medical, terminal or psychiatric illness (in patient or close family member) or no significantly impaired cognitive function.

## Sample size

This study aimed to randomise 105 men, 35 to each arm, in line with recent recommendations for pilot trials as sufficient to estimate key parameters for a full trial.[32] A sample size of 35 per arm is sufficient to allow the population variance to be estimated (eg, the SD in weight loss) with enough precision to deliver at least 80% power and 90% confidence in a full trial, with a standardised effect size between 0.2 and 0.5.[32]

## Participant recruitment

Participants were recruited via: (1) community outreach and (2) GP practice obesity registers.

### Community outreach

Community recruitment strategies that report success for recruiting men in disadvantaged areas were used.[11–13] This included researchers working on stands with study information leaflets and table banners in supermarkets, fitness centres, hospital foyers, health centres, council workplaces and community centres across both sites. Eleven of the 13 recruitment venues were within more disadvantaged areas based on the Scottish Index of Multiple Deprivation (SIMD quintiles 1 and 2).[33] Men who showed interest were given study information and asked to leave their contact details with researchers.

In addition, some men were recruited through word of mouth. This included researcher discussions with family, friends, colleagues and local workers who passed on study information to eligible men they knew. Information leaflets were distributed across localities in shops, libraries, barbers and community centres. Interested men then contacted researchers to get more information.

### GP practice obesity register letters

Practices within more disadvantaged areas (SIMD quintiles 1 or 2) were invited to participate in the study. Of the 33 practices invited (n=13 in site A, n=20 in site B), five practices participated (n=4 in site A, n=1 in site B).

Practice database searches identified potentially eligible men with a documented BMI of at least 30 kg/m$^2$. Lists were screened by clinical practice staff and details passed to the Health Informatics Centre at the University of Dundee who sent out GP-headed study invitation letters and information leaflets. Interested men contacted the research team or returned an 'opt-in' card.

### Baseline appointment

Men interested in participating were invited to attend a face-to-face appointment. Detailed study information was discussed, and written informed consent was provided. Anthropometric measurements were conducted and eligibility assessed. Participants then completed a baseline questionnaire and were randomised. Group-specific postrandomisation information was provided and discussed with the participant.

### Randomisation

At the baseline appointment, independent randomisation was performed by the researcher using the clinical trials unit's secure remote web-based system, stratifying by recruitment method (GP and community) and recruitment site (site A and site B).

### Blinding

The two recruitment researchers who also assessed outcomes and conducted qualitative interviews were mostly not blinded to group allocations. All other study team members were blinded. A demonstration of the feasibility for researcher outcome assessors to be blind to group allocation was conducted with 11 participants at the 6-month assessment. In this case, a researcher who had not previously met the participant arranged and undertook the assessment. These participants all complied with the request not to reveal their group allocation.

The trial statistician was fully blinded to intervention groups, and partially blinded for the control group due to the different response schedule (control participants were only assessed at baseline and 12 months to reflect 'usual life').

## Intervention components
### Narrative text messages

A narrative text message library consisting of 604 texts was written by a professional scriptwriter/researcher (MG) who designed the overall narrative with enough interlinked stories to engage participants over 12 months. Full details of the narrative texts development process are available elsewhere.[10] Narrative texts were sent to participants over the course of 12 months and were written from the point of view of a fictional character aiming to lose weight over 12 months. Narrative texts were written to appeal to men from disadvantaged backgrounds. All participants in the SMS+I and SMS only groups received automated texts according to a predetermined schedule. Participants could reply to texts but did not receive a response. Texts were sent between 08:00 and 22:00 and ranged from 0 to 5 texts per day depending on the requirements of the narrative approach used. All participants were scheduled to receive the same number of text messages. Some texts were personalised including participants' names and men could select whether weight information should be presented in kilograms or stones and pounds. No further personalisation and tailoring options were offered. Texts were delivered by the Health Information Centre in Dundee using existing automated technology linked to a clinical trials unit database.

### Endowment incentive

The incentive strategy was informed by existing evidence and men's preferences elicited using a Discrete Choice Experiment completed by 1045 men with obesity and reported elsewhere.[10] SMS+I participants were 'endowed' with a £400 incentive at baseline, which was placed into a hypothetical personal account at the University of

Stirling and given a mock-up personalised cheque. The full £400 could be secured by meeting weight loss targets at researcher assessments: 5% of body weight lost since baseline at 3 months (£50 secured/lost), 10% lost since baseline at 6 months (£150 secured/lost) and 10% lost since baseline at 12 months (£200 secured/lost) (see online supplementary 1). Weight loss was verified at all face-to-face appointments. At 6 and 12 months, men lost a proportion of the money for each % weight loss not attained between 5% and 10%. Weight at 12 months had to be less than at baseline to receive any money, regardless of whether interim weight loss targets had been met. Men received the money by direct bank transfer after the 12-month assessment. Feedback on meeting incentive targets was sent by automated text message and displayed on the personalised SMS+I webpage.

### Website
All trial participants were provided with a unique login ID for the Game of Stones website. The front page was accessible to all participants and included trial information and links to existing evidence-based online weight management resources. SMS+I and SMS only webpages had a brief biography and images of the fictional characters featured in narrative texts. Participants could enter their weight, pedometer steps, waist circumference and belt notches, which were displayed as basic visual progress charts. Only individual performance was displayed on the webpage, and no group averages were shown for social comparison. SMS+I webpages described the financial incentives and a visual progress chart of money secured/lost.

### Printed information
All participants received a weight loss fact sheet (British Dietetic Association, Weight Loss Food Fact Sheet) and a small card that could be carried in their wallet for noting website details and their appointment weight, weight loss targets and appointments.

### Pedometer
All participants received a study pedometer (3DFitBud, A420S, manufactured by 3DActive).

### Comparator group
At baseline, participants received access to the information section of the webpage, printed information (ie, a weight loss fact sheet) and a pedometer. They attended a baseline and 12-month appointment only. Control group participants were offered texts for 3 months commencing after the 12-month data collection point.

### Outcomes
The outcomes for this study related to whether the design of Game of Stones was both acceptable and feasible to deliver as a full-scale randomised controlled trial. An independent study steering committee advised whether the following prespecified progression criteria in the

study protocol were met sufficiently to proceed to a full trial.
1. Acceptability of the intervention and the control group (by the majority of the target group); willingness to be randomised.
2. Feasibility of recruiting 105 men in 4 months.
3. Twelve-month outcomes on at least 72% of men randomised per group, consistent with a recent UK weight management trial in men[34] and systematic reviews of male obesity literature.[4]
4. Evidence of mean weight loss of at least 3% of baseline weight at 12 months in any intervention group.
5. Commitment by, for example, government or National Health Service (NHS)/local authorities to fund the incentive intervention to ensure translation and sustainability.

### Outcome assessment
Outcomes were assessed at baseline, 3, 6 and 12 months for intervention participants (SMS+I and SMS only) and at baseline and 12 months for control participants. Individual appointments were at community centres, universities, NHS clinical research facilities, voluntary sector organisations, GP practice premises or the participant's home if no suitable alternative venue could be found. At the 12-month appointment, all participants received a £20 voucher as reimbursement for their time. No travel expenses were provided.

The text message delivery system automatically recorded the frequency of responses received to the texts. The website automatically recorded engagement with self-monitoring tools.

Self-report questionnaires were completed during appointments and measured sociodemographics, comorbidities, disability, ethnicity and perceptions on intervention acceptability. Overall satisfaction with the intervention was assessed with the item: '*On a scale from 0 (not satisfied at all) to 100 (completely satisfied). How satisfied are you with the Game of Stones programme?*'. Acceptability of the intervention was assessed with the stem '*Overall the Game of Stones programme has been…*' followed by options '*understandable*', '*useful*', '*helpful*', '*interesting*' and '*relevant*', with responses captured on a five-point scale ranging from '*totally disagree*' to '*totally agree*'. Helpfulness of the intervention was assessed with the item '*How helpful have you found the following in helping you lose weight?*' followed by options '*text messages*', '*website*' and '*pedometer*', with responses captured on a five-point scale ranging from '*totally unhelpful*' to '*totally helpful*'. Measures of overall satisfaction, acceptability and helpfulness were adapted from Dombrowski *et al.*[35]

Height was measured at baseline using a portable standing stadiometer (Seca 217, Birmingham, UK) to the nearest 0.1 cm. Prior to weight measurements, participants removed shoes and bulky clothing and items from their pockets. Weight was recorded using portable calibrated electronic scales (Marsden M420, Rotherham, UK) to the nearest 0.01 kg and waist circumference

using a tape measure (Seca 203, Birmingham, UK) to the nearest 0.1 cm at all assessment visits.

Information on possible adverse events was recorded at assessment visits via open questions and through automatic monitoring of text message replies including words like 'suicide', 'die', and 'death' indicative of potential adverse events, which were notified to the research team by email.

Any negative participant reactions to their randomised group were recorded in researcher field notes. Willingness to be randomised was assessed by recording the number of participants refusing randomisation.

Interviews took place during 3-month and 12-month appointments and were conducted face-to-face, and detailed methods are described elsewhere.[10] All audio recorded interviews were transcribed verbatim and anonymised. At 3 months, all SMS+I and SMS-only men were invited to participate in either a brief feedback or an in-depth interview (participant choice). The 3-month topic guide focused on the early acceptability of the intervention components in order to identify any refinements required over the remaining 9-month intervention. For the 12-month qualitative interviews, separate topic guides for the three trial groups were informed by the analysis of the 3-month interviews and researcher field notes to gain information power[36] to address the study objectives. Purposive sampling from the three trial groups was informed by the 3-month interview data and researcher field notes to provide diversity of perspectives. Researcher field notes taken at all assessments were referred to when interpreting interview data.

Qualitative interviews were conducted face-to-face with 50 of 58 men who attended the 3-month assessment (7–61 min, median=23 min). At 12 months, interviews were conducted with 14 participants from SMS+I (13–65 min, median=33 min), 13 from SMS only (10–59 min, median=28 min) and 6 from the control group (7–18 min, median=11 min). Fourteen interviews at 12 months lasted <20 min including control participants who had received no contact from the research team between baseline and 12 months.

## Analysis
### Quantitative analysis
All continuous variables were summarised and tabulated using the following descriptive statistics: N (number of valid non-missing responses), mean and SD. Likert-scale variables were treated as continuous measures. The frequency and percentages (based on the non-missing sample size) of observed levels are reported for all categorical measures. The proportion of individuals contacted who were recruited and the proportions retained and withdrawn at each assessment by group was determined. Missing weight data are presented as baseline observation carried forward (BOCF) and last observation carried forward (LOCF) in addition to observed cases only.

### Qualitative and mixed method data analysis
Anonymised transcripts of interviews were entered into NVivo12 software for analysis guided by the framework approach together with attributes for recruitment channel, attendance at assessments, trial group and participant characteristics and weight loss outcomes.[37] Reference to researcher field notes contributed to interview data interpretation. Iterative data collection and analysis were driven by the key feasibility and acceptability research questions and objectives. A coding frame was developed by three researchers independently reading a diverse sample of six interviews, followed by a team discussion to finalise the coding frame and identify key themes. Independent coding was conducted by four researchers (EC, NG, MM and RS) and checked for consistency by FH. Two independent researchers (EC and NG) who were not involved in any other aspect of the study assisted with coding and analysis to enhance rigour and reliability of data analysis. Emergent themes and interpretive analysis were discussed at weekly researcher meetings and at two coinvestigator qualitative interpretation meetings.

At the final stage of mixed methods data interpretation, the quantitative attributes on NVivo were drawn on to triangulate the analysis and suggest further avenues for interrogation. Matrix coding queries in NVivo were generated to cross-reference attributes for trial group, participant SIMD and weight loss outcome at 12 months, with nodes coded for views of incentives and texts messages, respectively. Credibility and reliability of qualitative analysis was enhanced by independent coding, use of memos to ensure transparency of interpretation and interpretive charting conducted with input from the wider study team.

## Patient and public involvement (PPI)
A continuous and responsive approach to PPI was adopted to prepare the grant application and throughout the study, as described by Gamble and colleagues.[38]

Continuous PPI was provided by a coinvestigator partnership with the Charities Men's Health Forum GB and Men's Health Forum in Ireland. The partnership commenced in 2011 with the Review Of MEn and Obesity (ROMEO) evidence syntheses of weight loss interventions for men with obesity.[4]

During the study, co-investigators from each of the Men's Health Forum charities attended trial management meetings to contribute to decisions, intervention development, data analysis, interpretation of findings and reporting. They provided feedback on the grant application, protocol, text messages, information materials and engaged wider involvement of men from their organisation to assist with appropriate language.[39]

Continuous PPI at the study oversight level was provided by two independent lay members of the study steering committee that met on three occasions.

Responsive PPI occurred at the study funding application stage. Co-investigator CG engaged men who provided PPI input in the Football Fans in Training (FFIT) Trial.[34] During the study, relevant members of the public were

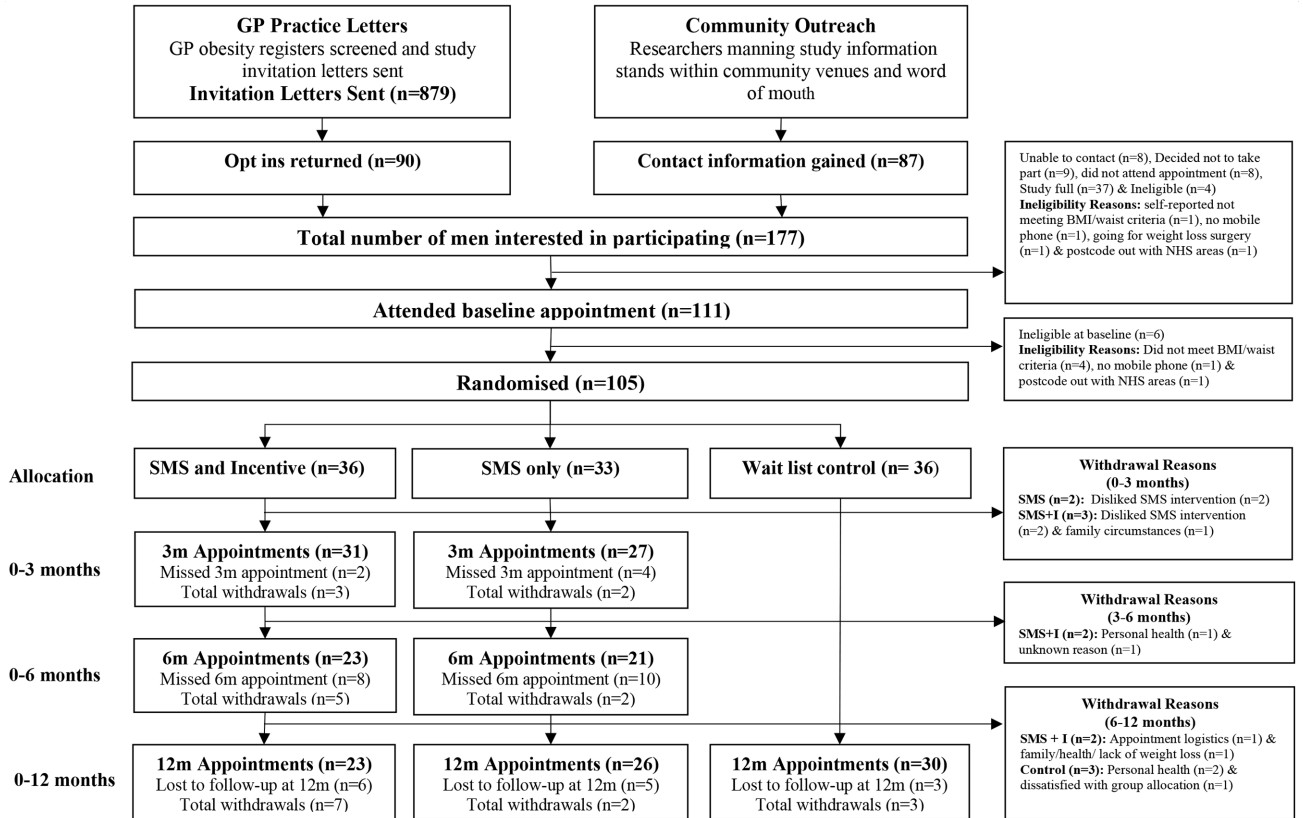

**Figure 1** CONSORT flow diagram. BMI, body mass index; CONSORT, Consolidated Standards of Reporting Trials; GP, general practitioner; NHS, National Health Service.

identified through several sources including: Men's Health Forum GB, Men's Health Forum in Ireland, Scottish Community Health Councils, Men's Sheds, University of Stirling PPI group, Alliance Scotland and other Co-investigator contacts. Effort was made to engage men from more disadvantaged communities. A total of 121 PPI contributors were involved in a range of study activities, including finding a study name, helping with wording of information material and helping with designing the intervention content. For full details, see the report to the funder.[10]

## RESULTS
### Recruitment and retention
Recruitment was completed in 4 months between March and June 2017 meeting the target number of 105 men randomised (figure 1).

Overall, 177 men expressed an interest in the study. Baseline appointments were attended by 111 (63%), of which 6 (5%) were ineligible. All 105 men were willing to be randomised and consented to be randomised to the SMS+I (n=36), SMS only (n=33) and control (n=36) groups.

GP practice recruitment was undertaken in five GP practices (site A: n=4, site B: n=1) with 45 men recruited and randomised through GP practices. The additional 60 men were recruited through community outreach.

The 12-month assessment was completed by 79/105 participants (75%). One of the 79 participants (control group) could not attend and provided self-reported information because he was out of the region due to work commitments, which means that 74% (78/105) of participants were retained for weight assessments at 12 months. Overall 12-month retention differed by group (SMS+I=23/36, 64%; SMS only=26/33, 79%; control=30/36, 83%). Fourteen participants (13%) were lost to follow-up, and 12 participants (11%) withdrew. Reasons for withdrawal were dislike of narrative texts (n=4), health (n=3), family (n=1), unknown (n=1), dissatisfaction with group allocation (n=1), appointment logistics (n=1) and multiple reasons (n=1).

### Baseline characteristics
Baseline characteristics are displayed in table 1. On average, participants were 52.2 (SD=13.1) years old, had a BMI of 35.7 (SD=5.9) and a waist circumference of 116.8 cm (SD=11.8). BMI ranged from 27.5 kg/m$^2$ to 62.5 kg/m$^2$. Twelve participants (11%) had a baseline BMI less than 30 kg/m$^2$ due to the additional weight circumference entry criterion of ≥40 inches (102 cm).

The majority of men lived in disadvantaged areas defined as SIMD quintiles 1 or 2 (n=62/104; 60%). Most participants were married (58%), reported at least one comorbidity (63%), were of white ethnicity (91%),

**Table 1** Participant baseline characteristics

| | SMS+I n=36 | SMS only n=33 | Control n=36 | Total n=105 |
|---|---|---|---|---|
| Age (years), mean (SD) | 50.9 (14.2)* | 52.5 (15.1)† | 53.1 (10.1) | 52.2 (13.1)‡ |
| Waist circumference (cm), mean SD | 115.8 (10.0) | 114.9 (12.7) | 119.5 (12.2) | 116.8 (11.8) |
| Weight (kg), mean (SD) | 108.6 (16.4) | 107.8 (20.2) | 110.7 (19.0) | 109.1 (18.4) |
| Height (cm), mean (SD) | 175.9 (6.6) | 175.2 (6.7) | 173.8 (5.9) | 175.0 (6.4) |
| BMI (kg/m$^2$), mean (SD) | 35.1 (5.3) | 35.1 (5.9) | 36.7 (6.5) | 35.7 (5.9) |
| BMI (kg/m$^2$) categories, n (%) | | | | |
| <30 | 5 (13.9) | 3 (9.1) | 3 (8.3) | 11 (10.5) |
| ≥30–>35 | 16 (44.4) | 19 (57.6) | 14 (38.9) | 49 (46.7) |
| ≥35–>40 | 10 (27.8) | 6 (18.2) | 7 (19.4) | 23 (21.8) |
| ≥40–>45 | 4 (11.1) | 2 (6.1) | 10 (27.8) | 16 (15.2) |
| ≥45–>50 | 0 (0.0) | 2 (6.1) | 1 (2. 8) | 3 (2.9) |
| ≥50 | 1 (2.8) | 1 (3.0) | 1 (2. 8) | 3 (2.9) |
| SIMD deprivation category, n (%) | | | | |
| SIMD 1 (most disadvantaged) | 11 (31.4)* | 12 (36.4) | 15 (41.7) | 38 (36.5)§ |
| SIMD 2 | 8 (22.9)* | 9 (27.3) | 7 (19.4) | 24 (23.1)§ |
| SIMD 3 | 6 (17.1)* | 3 (9.1) | 3 (8.3) | 12 (11.5)§ |
| SIMD 4 | 4 (11.4)* | 4 (12.1) | 6 (16.7) | 14 (13.5)§ |
| SIMD 5 (least disadvantaged) | 6 (17.5)* | 5 (15.2) | 5 (13.9) | 16 (15.4)§ |
| Marital status, n (%) | | | | |
| Cohabiting | 4 (11.1) | 2 (6.3) ¶ | 5 (13.8) | 11 (10.6)§ |
| Divorced | 2 (5.6) | 0 (0.0) ¶ | 2 (5.6) | 4 (3.9)§ |
| Married | 22 (61.1) | 21 (65.6)¶ | 17 (47.2) | 60 (57.7)§ |
| Separated | 1 (2.8) | 3 (9.4)¶ | 2 (5.6) | 6 (5.8)§ |
| Single | 7 (19.4) | 5 (15.6)¶ | 8 (22.2) | 20 (19.2)§ |
| Widowed | 0 (0) | 1 (3.1)¶ | 1 (2.8) | 2 (19.2)§ |
| Prefer not to say | 0 (0.0) | 0 (0.0)¶ | 1 (2.8) | 1 (1.0)§ |
| Comorbidities, n (%) | | | | |
| Arthritis | 11 (30.6) | 3 (9.1) | 8 (22.2) | 22 (21.0) |
| Cancer | 0 (0.0) | 0 (0.0) | 1 (2.8) | 1 (1.0) |
| Diabetes | 4 (11.1) | 5 (15.2) | 6 (16.7) | 15 (14.3) |
| Myocardial infarction | 2 (5.6) | 1 (3.0) | 0 (0.0) | 3 (2.9) |
| Hypertension | 7 (19.4) | 8 (24.2) | 7 (19.4) | 22 (21.0) |
| Stroke (including TIA) | 1 (2.8) | 1 (3.0) | 0 (0.0) | 2 (1.9) |
| Ethnic group, n (%) | | | | |
| Asian | 2 (5.6) | 1 (3.1)¶ | 1 (2.8) | 4 (3.8)§ |
| Black | 2 (5.6) | 1 (3.1)¶ | 1 (2.8) | 4 (3.8)§ |
| White | 32 (88.8) | 29 (90.7)¶ | 34 (94.4) | 95 (91.4)§ |
| Prefer not to say | 0 (0.0) | 1 (3.1) ¶ | 0 (0.0) | 1 (1.0)§ |
| Education, n (%) | | | | |
| Bachelor degree (=SVQ5) | 7 (19.4) | 6 (19.5)† | 11 (30.6) | 24 (23.3)** |
| HNC/HND (=SVQ4) | 4 (11.1) | 6 (19.5)† | 2 (5.6) | 12 (11.6)** |
| Higher grade/advanced higher/A-level or equivalent (=SVQ3) | 3 (8.3) | 5 (16.1)† | 1 (2.8) | 9 (8.7) ** |
| Masters/PhD or equivalent | 1 (2.8) | 1 (3.2)† | 3 (8.3) | 5 (4.8)** |

**Table 1** Continued

| | SMS+I n=36 | SMS only n=33 | Control n=36 | Total n=105 |
|---|---|---|---|---|
| No formal qualifications | 7 (19.4) | 3 (9.7)† | 10 (27.8) | 20 (19.4)** |
| Standard grade/GCSE/intermediate 1 or 2 | 6 (16.7) | 7 (22.6)† | 4 (11.1) | 17 (16.5)** |
| Still studying | 2 (5.6) | 1 (3.2)† | 3 (8.3) | 6 (5.8) ** |
| Vocational qualifications (=SVQ1+2) | 4 (11.1) | 1 (3.2)† | 0 (0.0) | 5 (4.8) ** |
| Other | 0 (0.0) | 1 (3.2)† | 0 (0.0) | 1 (1.0) ** |
| Prefer not to say | 2 (5.6) | 0 (0.0)† | 2 (5.6) | 4 (3.9) ** |
| Working status, n (%) | | | | |
| Full-time student | 4 (11.1) | 2 (6.3)¶ | 1 (2.8) | 7 (6.7) § |
| Employed – full time (30+ hours per week) | 16 (44.4) | 16 (50.0)¶ | 18 (50.0) | 50 (48.1) § |
| Employed – part time (8–29 hours per week) | 3 (8.3) | 3 (9.4)¶ | 0 (0.0) | 6 (5.8) § |
| Self-employed | 3 (8.3) | 3 (9.4)¶ | 1 (2.8) | 7 (6.7) § |
| Not in paid work | 2 (5.6) | 2 (6.2)¶ | 12 (33.3) | 16 (15.4) § |
| Retired | 8 (22.2) | 6 (18.7)¶ | 4 (11.1) | 18 (17.3) § |
| Household size, mean (SD) | 2.6 (1.3) | 2.6 (1.0) | 2.4 (1.4) | 2.5 (1.3) |

SIMD deprivation 1represents the most disadvantaged area, while quintile SIMD 5 represents the least disadvantaged area.
*N=35.
†N=31.
‡N=102.
§N=104.
¶N=32.
**N=103.
BMI, body mass index; GCSE, General Certificate of Secondary Education; HNC/HND, Higher National Diploma/Higher National Certificate; N, overall participants; n, participants within specific category; SIMD, Scottish Index for Multiple Deprivation; SVQ, Scottish Vocational Qualification.

reported having children (76%) and were in full-time employment (48%).

### Intervention fidelity

The intervention components were feasible to deliver with fidelity. The majority (95.4%) of the 38 214 text messages sent to the two intervention groups were delivered to the mobile phone, with 1782 (4.6%) having no delivery status (ie, lack of response from the mobile phone provider after 48 hours and invalid phone numbers). No major technical errors occurred. All participants who secured financial incentives were paid by direct bank transfer, in line with their stated preference.

### Acceptability of intervention components
#### Overall acceptability ratings and contamination

Table 2 displays intervention satisfaction indicators over the course of the study for participants attending assessments. Overall mean study satisfaction at 12 months was 81%, 77% and 87% for the SMS+I, SMS only and control groups, respectively. Satisfaction over time was comparable between both intervention groups. For scores on specific programme attributes, see table 2.

Helpfulness ratings of the narrative texts, webpage and pedometer were highest for the pedometer and relatively stable throughout the measurement time points. At 12 months, helpfulness ratings (out of 5) for the pedometer were 3.9, 4.0 and 3.7 for the SMS+I, SMS only and control groups, respectively. The narrative texts (SMS+I=3.4; SMS only=3.3) and the webpage (SMS+I=3.6; SMS only=3.4; control=3.5) were perceived as somewhat helpful on average at 12 months.

Minimal contamination between intervention groups was observed, with one participant at 6 and 12 months (SMS+I group) and another at 12 months (control group) reporting meeting other men in the study.

#### Acceptability of narrative text messages

Some men sent spontaneous replies to the narrative texts (0–3 months n=25/69, 36%; 3–6 months n=8/69, 12%, 6–12 months n=13/69, 19%; see online supplementary 2). Most spontaneous replies were received between 0 and 3 months (n=370 replies) and decreased at 3–6 months (n=16 replies) and 6–12 months (n=39 replies).

Eleven participants (11/69, 16%) requested to no longer receive texts but asked to remain in the trial. Requests to stop narrative texts were similar in both groups (SMS+I=5/36, 14%; SMS only=6/33, 18%) and occurred throughout the study. Four participants (4/69, 6%) withdrew due to dislike of narrative texts between baseline and 3 months, two in each intervention group.

Qualitative interviews at 3 and 12 months demonstrated varied views on the narrative texts. Participants' views

**Table 2** Programme satisfaction and contamination at 3, 6 and 12 months

| | 3 months | | 6 months | | 12 months | | |
| --- | --- | --- | --- | --- | --- | --- | --- |
| | SMS+I (n=31) | SMS only (n=27) | SMS+I (n=23) | SMS only (n=21) | SMS+I (n=22) | SMS only (n=26) | Control (n=30) |
| Programme satisfaction (0–100), mean (SD) | 80.3 (21.1) | 75.0 (22.2) | 76.2 (29.6) | 79.0 (20.7) | 80.9 (20.0) | 77.0 (20.8) | 87.3 (17.5)* |
| Programme has been… (1=low, 5=high), mean (SD) | | | | | | | |
| Understandable | 4.5 (0.9) | 4.4 (0.8)† | 4.4 (1.2)‡ | 4.4 (1.2) | 4.6 (1.1)§ | 4.6 (1.1)¶ | 4.6 (0.9)* |
| Useful | 4.4 (0.9) | 4.0 (1.0) | 4.1 (1.3) | 4.1 (1.1)§ | 4.1 (1.2) | 4.0 (1.0) | 4.3 (0.9) |
| Helpful | 4.3 (0.9)** | 3.9 (1.0) | 4.1 (1.3) | 4.1 (1.1)§ | 4.1 (1.10) | 4.0 (1.0) | 4.3 (0.9) |
| Interesting | 4.3 (1.0) | 4.2 (1.1) | 4.2 (1.1) | 4.4 (1.3)§ | 4.3 (0.8) | 4.3 (1.2) | 4.5 (1.0) |
| Relevant | 4.2 (1.1) | 3.9 (1.2) | 4.0 (1.3) | 4.3 (1.1)§ | 4.1 (1.1) | 4.2 (1.1) | 4.5 (1.0) |
| Helpfulness (1=low, 5=high), mean (SD) | | | | | | | |
| Text messages | 3.4 (1.5) | 3.1 (1.3) | 3.1 (1.3) | 3.6 (1.3) | 3.4 (1.3) | 3.3 (1.4) | n/a |
| Website | 3.3 (1.2)†† | 3.3 (0.9)† | 3.3 (1.2) | 3.2 (0.8)§ | 3.6 (1.0) | 3.4 (0.7)‡‡ | 3.5 (1.1)† |
| Pedometer | 4.1 (1.2)** | 4.1 (0.9) | 3.7 (1.4) | 4.0 (1.0) | 3.9 (1.2)§§ | 4.0 (1.0)¶ | 3.7, (1.4) |
| Met other men in programme, n (%) | | | | | | | |
| Yes | 0 (0.0) | 0 (0.0) | 1 (4.3) | 0 (0.0) | 1 (4.5) | 0 (0.0) | 1 (3.3) |

*N=29.
†N=26.
‡N=22.
§N=20.
¶N=25.
**N=30.
††N=28.
‡‡N=24.
§§N=21

ranged from positive to indifferent to negative. Those who liked the narrative texts found the storyline entertaining, engaging and some participants felt a certain camaraderie for the main character:

> You get involved in the content and you start following the script, they're very funny and me being a (from city name) you can see the funny side and you can see the wit in that, …. and that's what keeps you reading them. (210010, SMS only, 12 months)

For other participants, the frequency of texts became a source of irritation, the storyline did not resonate with their own experience nor could they empathise with the fictional characters:

> I'm not a lover of chocolate, biscuits, cake, ice cream, I hardly ever touch them, hardly ever, so hearing about somebody eating pizza or sweets means nothing to me because I don't eat them anyway. (120002, SMS+I, 12 months)

Some were uncertain of the role of the texts in helping to support weight loss. Others expressed an indifference towards the texts and a decreasing interest in the story over time leading to infrequent engagement with text content. However, the regular reminder of being a participant in a weight loss programme through receiving texts was seen as important:

> I often wonder, do they relate to myself here. And the only commonality is both trying to lose weight. So in that respect it's just a reminder all the time, you should be reviewing your weight and watching what you're eating, so… from that it's a positive, yeah. (220017, SMS+I, 3 months)

A matrix coding query and analysis of the quotations generated from the NVivo qualitative and demographic data suggested no obvious relationships between the participant SIMD level and the qualitative accounts of acceptability of the narrative texts. This is in line with quantitative data showing similar acceptability ratings for overall programme satisfaction and helpfulness ratings of the narrative texts across SIMD levels (see online supplementary 3). Some men would have liked more interactivity built into the texts; however, the text message system was set up for unidirectional messaging only.

### Acceptability of endowment incentive

At 3 months, 31/36 SMS+I participants attended the appointment, with eight men losing 5% or more of their baseline weight and securing £50. Attendance at 6 months was lower (n=23/36), with five participants achieving the weight loss target of >10% weight loss securing £150 each, and a further three participants losing between ≥5% and 10% of weight securing between £75 and £150. At 12

months, 23/36 participants attended the appointment, with three achieving >10% weight loss securing £200, and a further seven losing between ≥5% and 10% of weight securing between £100 and £200.

Overall, £2955 was paid in total to the 11 participants (11/36, 31%) successfully meeting or partially meeting weight loss targets. The full £400 was secured by three participants (3/36, 8%). The cost of the incentive per participant was £81.94 (95% CI £34.59 to £129.30). All participants who completed the study and secured money at 3 or 6 months received it at the end of the study. No participant lost previously secured money due to weight regain to baseline weight at 12 months. One participant secured money at an early appointment but withdrew from the study, did not attend at 12 months and did not receive any money.

No negative views were expressed about the incentives in qualitative interviews. Only one man reported being motivated by the financial incentive, and many expressed indifference. Participants reported the weight loss and health benefits as sufficiently rewarding.

The money's not that much of an issue. I think the incentive for me is the health that, at the end of the day, your health improves. That's the most important incentive. The money, it's fine, it's there, but it's not the incentive. (120022, SMS+I, 3 months)

Some interpreted the incentives as a final reward linked to their weight loss targets, whereas others fully understood the intended use of the loss aversion concept:

[W]e all know how much more distress it causes you to lose money than the pleasure of finding a fiver down the back of the couch, you know, so I think if you lose something it almost has a bigger impact on you than if you gain. (220040, SMS+I, 3 months)

### Acceptability of study webpages and pedometer

Some participants used the self-monitoring features on the webpage (see online supplementary 2) and reported valuing the ability to visually track their weight in interviews. However, most did not access the website. The majority (77%) did not own a pedometer at baseline, and pedometers were highly acceptable to participants.

### Acceptability of control group

Most control group participants (30/36, 83%) attended the 12-month appointment. Six qualitative interviews were conducted with men in the control group covering topics including acceptability of waiting 12 months for text messages, views about the study and improvement suggestions. Many reported being pleased to be involved in a research study. One participant had a strong negative reaction to being randomised to the control group and subsequently withdrew when invited to the 12-month assessment.

### Weight outcomes

Table 3 displays weight change outcomes at 3, 6 and 12 months. The difference in mean percentage weight loss at 12 months for observed cases was −2.51% (95% CI −6.03% to 1.01%) for SMS+I versus control and −0.51% (95% CI −3.91% to 2.89%) for SMS only versus control. The difference in absolute weight loss at 12 months was −2.87 kg (95% CI −6.82 to 1.08) for SMS+I versu control and −0.57 kg (95% CI −4.40 to 3.25) for SMS only versus control.

The SMS+I group displayed mean weight loss of over 3% at 12 months (−3.51%, SD=5.83). The SMS only and control groups remained below 3% weight loss on average (SMS only=−1.51%, 4.65; control=−1.00%, SD=5.31). The highest mean percentage weight loss at 12 months for BOCF was −2.24% (SD=4.93) in the SMS+I group, followed by −1.19% (SD=4.16) and −0.80% (SD=4.77) in the SMS only and control groups, respectively.

### Harms and unintended consequences

No harms or unintended consequences were reported.

### Progression to full trial

An independent study steering committee agreed that the Game of Stones study had demonstrated acceptability and feasibility agreeing that overall the prespecified progression criteria were sufficiently met to support a full three-arm multisite randomised controlled trial (RCT).

## DISCUSSION
### Principal findings

This study successfully recruited 105 men from across the socioeconomic spectrum to a three-armed RCT and overall achieved 74% retention at 12-months follow-up. Men living in more disadvantaged areas formed 60% of the sample. Narrative texts were broadly acceptable, but some participants disengaged or withdrew from the study due to dislike of texts. The endowment incentives were acceptable, and improving health was reported as a key motivator for weight loss. Positive indicative effects of weight loss were found in intervention and control groups.

### Strengths and weaknesses of the study

This study was underpinned by the ROMEO systematic reviews and qualitative evidence synthesis of weight loss interventions for men with obesity,[40 41] with a continuing PPI partnership with the Men's Health Forum GB and Ireland charities. Mixed methods research combined quantitative and qualitative data which, together with patient, public and stakeholder involvement, provided a multilens perspective on this study.

Most assessments were not blind to group allocation. The difference in assessment schedules for intervention and control groups meant that study staff were only partially blind. A full trial should ensure full blinding of all outcome assessments. Researchers were unable

**Table 3** Weight change at 3, 6 and 12 months

| | 3 months | | 6 months | | 12 months | | |
|---|---|---|---|---|---|---|---|
| | SMS+I n=36 | SMS only n=33 | SMS+I n=36 | SMS only n=33 | SMS+I n=36 | SMS only n=33 | Control n=36 |
| **Weight change (kg), mean (SD)** | | | | | | | |
| Observed cases only | −2.79 (3.50)* | −1.97 (3.97)† | −4.59 (5.62)‡ | −3.30 (4.92)§ | −3.93 (5.74)‡ | −1.64 (5.64)† | −1.06 (6.29)¶ |
| BOCF | −2.40 (3.38) | −1.55 (3.60) | −2.93 (4.98) | −2.10 (4.21) | −2.51 (4.94) | −1.29 (5.03) | −0.86 (5.64) |
| LOCF | −2.40 (3.38) | −1.86 (3.88) | −3.38 (4.92) | −2.21 (4.39) | −2.98 (4.91) | −1.33 (5.04) | −0.86 (5.64) |
| **Weight change (%), mean (SD)** | | | | | | | |
| Observed cases only | −2.54 (3.47)* | −1.95 (3.72)† | −4.20 (5.54)‡ | −3.02 (4.22)§ | −3.51 (5.83)‡ | −1.51 (4.65)† | −1.00 (5.31)¶ |
| BOCF | −2.18 (3.33) | −1.53 (3.39) | −2.69 (4.84) | −1.92 (3.65) | −2.24 (4.93) | −1.19 (4.16) | −0.80 (4.77) |
| LOCF | −2.18 (3.33) | −1.53 (3.39) | −3.11 (4.80) | −2.06 (3.85) | −2.68 (4.92) | −1.22 (4.16) | −0.80 (4.77) |
| **Weight change categories (observed cases only), n (%)** | | | | | | | |
| Weight gain | 6 (19.4)* | 8 (30.8)† | 2 (8.7)‡ | 6 (28.6)§ | 5 (21.7)‡ | 10 (38.5)† | 14 (48.3)¶ |
| 0–<3% weight loss | 13 (41.9)* | 10 (38.5)† | 10 (43.5)‡ | 7 (33.3)§ | 5 (21.7)‡ | 9 (34.6)† | 7 (24.1)¶ |
| ≥3–<5% weight loss | 4 (12.9)* | 4 (15.4)† | 3 (13.0)‡ | 2 (9.5)§ | 3 (13.0)‡ | 4 (15.4)† | 0 (0.0)¶ |
| ≥5–<10% weight loss | 8 (25.8)* | 2 (7.7)† | 3 (13.0)‡ | 4 (19.1)§ | 7 (30.4)‡ | 2 (7.7)† | 6 (20.7)¶ |
| ≥10% weight loss | 0 (0.0)* | 2 (7.7)† | 5 (21.7)‡ | 2 (9.5)§ | 3 (13.0)‡ | 1 (3.9)† | 2 (6.9) |
| **BMI change, mean (SD)** | | | | | | | |
| Observed cases only | −0.89 (1.1)* | −0.73 (1.32)† | −1.45 (1.79)‡ | −1.04 (1.49)§ | −1.24 (1.89)‡ | −0.49 (1.68)† | −0.47 (2.13)¶ |
| BOCF | −0.75 (1.07) | −0.59 (1.22) | −0.92 (1.58) | −0.66 (1.28) | −0.78 (1.60) | −0.37 (1.50) | −0.37 (1.96) |
| LOCF | −0.75 (1.07) | −0.59 (1.22) | −0.93 (1.56) | −0.69 (1.34) | −0.93 (1.59) | −0.38 (1.50) | −0.37 (1.96) |
| **Waist circumference change (cm), mean (SD)** | | | | | | | |
| Observed cases only | −3.74 (4.62)* | −3.14 (4.3)† | −4.70 (6.29)‡ | −3.14 (4.34)§ | −4.40 (6.08)‡ | −2.30 (4.37)† | −2.26 (4.97)¶ |
| BOCF | −3.22 (4.47) | −2.49 (3.9) | −3.00 (5.49) | −2.00 (3.76) | −2.81 (5.82) | −1.82 (3.98) | −1.82 (4.53) |
| LOCF | −3.22 (4.47) | −2.49 (3.9) | −3.71 (5.76) | −2.23 (4.03) | −3.51 (4.48) | −1.85 (3.97) | −1.82 (4.53) |

*N=31.
†N=26.
‡N=23.
§N=21.
BMI, body mass index; BOCF, baseline observation carried forward; LOCF, last observation carried forward.

to collect data via qualitative interviews, questionnaires and anthropometric measures with men who withdrew from the study or were lost to follow-up at the 12-month appointment. Those providing a withdrawal reason may have provided socially desirable responses. Text messages were personalised by including the participant's name and weight unit preference, but no tailoring or interactivity was possible due to technical limitations of the delivery system. This feasibility trial was not powered to detect effects on weight loss, and in line with recommendations,[27] no p values are reported. Weight outcomes should be interpreted with caution.

### Relation to other studies

The Game of Stones study adds to the evidence base demonstrating the feasibility of recruiting men for research on sensitive subjects, such as obesity, through community outreach and GP practice lists.[11–13]

The average BMI and age of the study participants of around 35 kg/m$^2$ and 50–55 years is similar to UK weight management trials recruiting in the community,[34] primary care[42] or a combination of community and primary care.[12] Three international text message-based weight management studies examining outcomes after 12 months recruited younger participants with lower BMI.[43–45] However, three mixed sex weight management studies with financial incentives (two including text message components) reported broadly similar participant demographics to this study.[46–48]

Retention levels of 74% were acceptable and are similar to systematic review evidence of men-only weight loss interventions, which found an average retention of 78%, ranging from 44% to 100%.[41] International text message studies with mixed sex participants report similar retention rates at 12 months of 70.3%[44] and 73%,[43] with one study conducted in Latvia reporting 93% retention.[45] Two large UK studies recruiting mixed sex participants with obesity to weight management in primary care reported 75%[42] and 81%[49] retention at 12 months.

Two previous narrative text message intervention studies in men targeting alcohol reduction reported high acceptability levels and no negative reactions.[12 13] However, these studies were only 12 weeks in duration and overall contained fewer texts. The narrative texts in this study were broadly acceptable, although some negative reactions were reported. Study withdrawal due to texts and requests to stop texts suggests that the current narrative texts may not be universally acceptable to men. The text messages were designed by a professional scriptwriter/researcher with PPI input from the target population of men from disadvantaged backgrounds, but the narrative texts were similarly acceptable across the socioeconomic spectrum. Intervention fidelity was high with the majority of sent texts successfully delivered, and no adverse events were encountered, similar to previous text message-based intervention studies.[11–13]

The use of a financial incentive strategy for weight loss in men with obesity across the socioeconomic spectrum was feasible and acceptable. Two previous incentive studies used text messages to inform participants about their incentive achievement of weight loss targets and found it acceptable.[46 47] This extends the evidence base on the use of financial incentives as a complementary behaviour change strategy alongside other components.[22 50]

The financial incentive strategy was designed with future sustainability in mind and to be attractive to public sector funders. The text messages and incentive components are mostly automated, encourage self-management of weight and have a low administrative burden with one bank transfer pay out after 12 months and verification of weight loss at 3, 6 and 12 months. Yancy et al similarly provided incentives at 3 and 6 months for their 6-months incentive intervention.[46] John et al asked participants to be weighed monthly, for which they received $20 per visit.[47] Few RCTs examine financial incentive strategies that are delivered for at least 12 months.[50] Two other studies providing financial incentives for 12 months provided an intensive financial incentive schedule either weekly[51] or monthly.[52] Previous studies typically provided participants with weekly weight loss goals such as 1 lb (0.45 kg) per week, on which financial incentives were contingent.[50] While these goals are similar to the weight loss targets in the current study, there were no intermittent or weekly targets. A balance needs to be struck between having more regular weight measurements and more immediate pay-outs, costs and future sustainability.

Patel et al[26] applied a similar framing which they called a loss incentive. University employees were allocated a monthly hypothetical incentive of $42 upfront, and $1.40 was taken away each time their daily goal of 7000 steps was not met. The current study adds to the evidence base demonstrating the acceptability and feasibility of loss framed interventions for weight loss and in settings outside workplaces.

### CONCLUSION

This study tests a novel combination of narrative SMS with endowment incentives that have not previously been combined in this way to address weight loss and weight loss maintenance in men with obesity. This three-arm feasibility trial recruited men from across the socioeconomic spectrum with the majority coming from disadvantaged areas, had an acceptable retention rate and was broadly acceptable to most participants, and feasible to deliver. Acceptability and feasibility progression criteria were met. A full trial is warranted to determine effectiveness and cost-effectiveness, with consideration of scalability.

**Author affiliations**
[1]Department of Kinesiology, University of New Brunswick Fredericton, Fredericton, New Brunswick, Canada
[2]Division of Psychology, University of Stirling, Stirling, UK
[3]Nursing, Midwifery and Allied Health Professional Research Unit, University of Stirling, Stirling, UK
[4]Health Economics Research Unit, University of Aberdeen, Aberdeen, UK

[5]Institute for Health Research and Innovation, University of the Highlands and Islands, Inverness, UK
[6]Health Services Research Unit, University of Aberdeen, Aberdeen, UK
[7]Mens Health Forum in Ireland, Dublin, Ireland
[8]Nursing, Midwifery and Allied Health Professions Research Unit, Glasgow Caledonian University, Glasgow, UK
[9]Institute of Health and Wellbeing, University of Glasgow, Glasgow, UK
[10]Tayside Clinical Trials Unit, University of Dundee, Dundee, UK
[11]Health Informatics Centre, University of Dundee, Dundee, UK
[12]Centre for Public Health, Queen's University Belfast, Belfast, UK
[13]Men's Health Forum, London, UK

**Acknowledgements** We would like to acknowledge the following people who have contributed to this study: all men who contributed to patient and public involvement (PPI) and who participated in the study. Colin Fowler from Men's Health Forum in Ireland, who was involved in the ROMEO systematic review that informed this study, commented on the grant application, protocol and final report. Together with PC, they accessed wider PPI in Ireland. FH, Director, Tayside Clinical Trials Unit, oversaw randomisation, data management and the running of the study from a Clinical Trials Unit perspective. Hannah Collacott, who was employed as a research assistant at the Health Economics Research Unit, University of Aberdeen in 2016-2017. She undertook the think aloud interviews, PPI and assisted in the design and preliminary analysis of the Discrete choice experiment (DCE) described in the final report to the funder. (Dombrowski *et al*, in submission). Clinical, Community and Academic Networks who advised our team on many aspects of this study, including: NHS Research Scotland Primary Care Network and the general practitioner practices involved, Alliance Scotland, Community Health Councils and Health Centres. The following postgraduate masters students who accompanied the research assistants when recruiting men in the community in alphabetical order: Norelle Calder-Macphee, Ines Elenin Lucy Gunatillake, Bryony Lang, Heather Renwick and Amy Spiteri. Karen Stanton, Lorna Kerr, Karen Murray, Siobhan McDermott and Suzannah Hunter who provided administrative support. The prior work of Professor Crombie and his team at Dundee who developed the software to deliver the SMS and assisted with the grant application for this study (Crombie *et al* 2017; Crombie *et al* 2018; Irvine *et al* 2017). The Study Steering Committee: Associate Professor Ed Juszczak, Director, NPEU Clinical Trials Unit, Nuffield Department of Population Health, University of Oxford (Chair). Professor Kate Jolly, Professor of Public Health & Primary Care, University of Birmingham. Mr Mark Kelvin, Programme Director (until 2018), Health and Social Care Alliance Scotland (the ALLIANCE). Mr Andrew Shanahan, Director MAN v FAT – the weight loss website for men. Ms Martine Stead, Deputy Director, Institute for Social Marketing, University of Stirling Miss Joyce Thompson Dietetic Consultant in Public Health Nutrition and Tayside Nutrition MCN Lead Clinician, NHS Tayside.

**Contributors** SUD co-led the study with PH as joint chief investigator and co-led the design and overall conduct of the study and led the writing of the manuscript, overseen by PH. MM undertook the study set up, recruitment and data collection, coordinated various PPI activities and contributed substantially to qualitative and quantitative data analysis and interpretation. MvdP was a coinvestigator, contributed to the design and the interpretation of findings and led the DCE and health economic analysis. MG was a coinvestigator, contributed to the design and the interpretation of findings and led the development of the narrative SMS component, and the related PPI. AA was a coinvestigator and contributed to the design and the interpretation of findings. PC and MT linked to PPI networks and contributed to the design and interpretation of the findings. AE was a coinvestigator, contributed to the design and the interpretation of findings and oversaw the statistical analysis. NG and EC contributed to the analysis and interpretation of the qualitative data. CG was a coinvestigator, facilitated PPI and contributed to the design and interpretation of findings. FH was a coinvestigator, oversaw the qualitative data collection and analysis and contributed to the interpretation of findings. AH undertook the statistical analysis at Tayside Clinical Trials Unit. CJ was a coinvestigator, contributed to the design and the interpretation of findings and oversaw the automated SMS delivery system, recruitment tracking, the study website and integration of data with databases at Tayside Clinical Trials Unit. FK and MCM were coinvestigators and contributed to the design and the interpretation of findings. RS contributed to recruitment, data collection and undertaking qualitative interviews and contributed to the interpretation of findings.

**Funding** This report presents independent research commissioned by the NIHR. The views and opinions expressed by authors in this publication are those of the authors and do not necessarily reflect those of the NHS, the NIHR, MRC, CCF, NETSCC, the Public Health Research programme or the Department of Health. The views and opinions expressed by the interviewees in this publication are those of the interviewees and do not necessarily reflect those of the authors, those of the NHS, the NIHR, MRC, CCF, NETSCC, the Public Health Research programme or the Department of Health.The Nursing Midwifery and Allied Health Professionals Research Unit, the Health Services Research Unit and Health Economics Research Unit are funded by the Chief Scientist Office of the Scottish Government Health and Social Care Directorate.

**Competing interests** MG is director of Eos Digital Health Ltd, which holds all rights to DNA, The Digital Narrative Approach, deployed to design and write the narrative text intervention.

**Patient consent for publication** Not required.

**Provenance and peer review** Not commissioned; externally peer reviewed.

**Data availability statement** Data are available on reasonable request. Access to data can be arranged through the coprincipal investigators of the study: Professor Pat Hoddinott (University of Stirling, p.m.hoddinott@stir.ac.uk) and Dr Stephan Dombrowski (University of New Brunswick, stephan.dombrowski@unb.ca) to discuss data sharing, data requirements and conflicts of interest, in line with any EU and other regulations, including ethics approvals.

**ORCID iDs**
Stephan U Dombrowski http://orcid.org/0000-0001-9832-2777
Pat Hoddinott http://orcid.org/0000-0002-4372-9681

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
