## [Reviewer comments · BMJ Open]

ARTICLE DETAILS

TITLE (PROVISIONAL)	Game of Stones: Feasibility randomised controlled trial of how to engage men with obesity in text message and incentive interventions for weight loss
AUTHORS	Dombrowski, Stephan U; McDonald, Matthew; van der Pol, Marjon; Grindle, Mark; Avenell, Alison; Carroll, Paula; Calveley, Eileen; Elders, A; Glennie, Nicola; Gray, Cindy; Harris, Fiona; Hapca, Adrian; Jones, Claire; Kee, Frank; McKinley, Michelle C.; Skinner, Rebecca; Tod, Martin; Hoddinott, Pat

VERSION 1 - REVIEW

REVIEWER	Sohye Lee The University of Memphis, United States
REVIEW RETURNED	09-Oct-2019

GENERAL COMMENTS	Thank you so much for allowing me to review this manuscript. I have made several points for potential revision for authors. Introduction 1. The background section provided significance, background, and problem of research. The literature review was up-to-date and clear. However, it would have been strengthened by adding more clear points about what is known and unknown area of this research. You mentioned that there are no text message intervention studies for men only. How about the findings from financial incentive interventions for underserved populations (e.g., low-income families)?2. The objectives are clearly defined but not clearly linked with the study findings. How did you measure their willingness to be randomized or the feasibility of recruiting? The research questions should be clear, and all the reports should be congruent with the research questions. Methods 1. Based on your methods and results sections, the mixed-methods design was used in this study. I think the research design should be clearly stated with rationales.2. The methods section could be tightened, focusing on your research questions.3. The setting can be explained more in detail. Could you explain more about the high level of disadvantage? How did you define the level of disadvantage? Also, you should use the complete sentence for the body of the main text.
---

	Results  1. Tables 1 and 3: Did you detect any significant differences between groups (SMS+I vs. SMS only) at baseline, 3-, 6-, and 12-month? The statistical analyses are somewhat brief. 2. It would be strengthened by organizing the results based on their research objectives/questions. Discussion  1. What kinds of limitations of this study did you have? The weaknesses were discussed, but it would have strengthened by adding more aspects of some threats of internal and external validity issues for this study. 2. Also, it would have enhanced by adding implications for practice and future research. I didn't review the study protocol. This manuscript is well-written, and the topic is significant and interesting. I believe the authors can improve the quality of the report. Thank you.
--	--

REVIEWER	Robert L Newton, Jr. Pennington Biomedical Research Center, USA
REVIEW RETURNED	16-Oct-2019

GENERAL COMMENTS	The study reports on the use of narrative text messages and financial incentives for weight loss in men. The use of financial incentives in continuing to gain attention and the study utilizes a novel form of text messaging in this space. While this is a novel investigation, there are several issues that impede its ability to contribute to the current literature base.  1. Having assessment staff who are not blinded is a significant threat to internal validity. It is clear that blinding was possible but it is not clear why it was not part of the procedures. 2. The lack of assessment of control participants at 6 months effectively unblinded all staff to the control participants. This is another methodological issue that threatens internal validity. 3. It is not clear what determined the frequency of text messages. 4. The website seems to be a component of the intervention. It does not appear to be a part of the inclusion criteria and it is not used to assess feasibility. 5. Why were control participants interviewed? I realize that detailed information on the interviews is provided in another manuscript, however, if these interviews are relevant for feasibility and acceptability, more information on the interview questions are needed. 6. There are no predefined standards for feasibility or acceptability. Therefore, it is difficult to determine if the study was feasible or acceptable. 7. Lines 18-22 and 28-31 on page 18 appear to be contradictory. 8. The authors need to discuss what steps are next given their interpretation of the findings. Where does this study lead them? What are future steps for other researchers in this space? What does this study tell us?
--

VERSION 1 – AUTHOR RESPONSE

Reviewer: 1

#	Comment	Response
1	The background section provided significance, background, and problem of research. The literature review was up-to-date and clear. However, it would have been strengthened by adding more clear points about what is known and unknown area of this research. You mentioned that there are no text message intervention studies for men only. How about the findings from financial incentive interventions for underserved populations (e.g., low-income families)?	Thank you for this suggestion. We have now extended the background information on financial incentives for weight loss in underserved populations. We have added an additional systematic review that underlines the potential for financial incentives to support behaviour change in low-income adults. Reference. Haff N, Patel MS, Lim R, et al. The role of behavioral economic incentive design and demographic characteristics in financial incentive-based approaches to changing health behaviors: A meta-analysis. Am J Health Promot 2015;29(5):314-23. doi: 10.4278/ajhp.140714-LIT-333 We have added a sentence outlining that little evidence exists that focuses on using financial incentives to support weight management in individuals from disadvantaged backgrounds, particularly for men. We support this statement by citing a study using financial incentives for weight loss in women from disadvantaged backgrounds (Leahey et al., 2017), a recent protocol published in BMJ Open for a study examining financial incentives for weight loss in mixed sex low-income patients (Jay et al., 2019), and the ROMEO review which provides a comprehensive summary of the available literature on weight management in men, finding few incentive-based studies in men (Robertson et a., 2014). References. Jay M, Orstad SL, Wali S, et al. Goal-directed versus outcome-based financial incentives for weight loss among low-income patients with obesity: Rationale and design of the Financial Incentives for Weight Reduction (FIReWoRk) randomised controlled trial. BMJ Open 2019;9(4) doi: 10.1136/bmjopen-2018-025278 Leahey TM, LaRose JG, Mitchell MS, et al. Small Incentives Improve Weight Loss in Women From Disadvantaged Backgrounds. American Journal of Preventive Medicine 2018;54(3):e41-e47. doi: 10.1016/j.amepre.2017.11.007

		Robertson C, Archibald D, Avenell A, et al. Systematic reviews of and integrated report on the quantitative, qualitative and economic evidence base for the management of obesity in men. HTA 2014;18(35):1-424. The introduction now reads: “Systematic review evidence of financial incentives for behaviour change highlights the potential for incentives to change behaviours in low-income adults¹⁴ and help reduce health inequalities.¹⁵ However, little evidence exists that focuses on using financial incentives to support weight management in individuals from disadvantaged backgrounds^{16 17}, particularly for men⁴ⁿ, p.6.
2	The objectives are clearly defined but not clearly linked with the study findings. How did you measure their willingness to be randomized or the feasibility of recruiting? The research questions should be clear, and all the reports should be congruent with the research questions.	Thank you for this comment. We have previously tried to report results by research objectives when writing the report to the study funders. However, the results were less succinct than the current reporting style, and it led to unnecessary repetition of information. We prefer to present study outcomes in their current order. This study employed mixed methods with extensive public and stakeholder involvement. Data collection and analysis used the principle of triangulation to integrate findings across the entire study to answer the feasibility objectives (O’Cathain et al., 2010). These objectives relate to the overall research aim as stated in the manuscript (p.7): “to examine the acceptability and feasibility of a men-only weight management intervention consisting of narrative text messages, with and without an endowment incentive, compared to waiting list control.” Regarding willingness to be randomised, we have added additional information on how we assessed this in the Outcome Assessment section, p.13: “Any negative participant reactions to their randomised group were recorded in researcher field notes to assess willingness to be randomised.” Regarding feasibility of recruitment, we now include the pre-specified progression criteria which state the criterion used for judging feasibility of recruitment: “Feasibility of recruiting 105 men in four months.”, p.12. Reference.

		O'Cathain A, Murphy E, Nicholl J. Three techniques for integrating data in mixed methods studies. BMJ . 2010;341:c4587.
3	Based on your methods and results sections, the mixed-methods design was used in this study. I think the research design should be clearly stated with rationales.	Thank you for this suggestion. We have added additional information on study design and the rationale on p. 7 under the heading of "Trial design". This section now reads: "A three-arm individually randomised parallel-group controlled feasibility trial was with an integrated qualitative and quantitative mixed methods approach²⁷. Informed by MRC guidance on the evaluation of complex interventions, the study included an integrated mixed methods process evaluation^{28 29}. Drawing on both qualitative and quantitative approaches allows the team to explore participants views (acceptability) of the intervention with participants as well as explore implementation processes such as recruitment, retention and barriers and facilitators to these." References. O'Cathain A, Murphy E, Nicholl J. Three techniques for integrating data in mixed methods studies. BMJ. 2010;341:c4587. Craig, P., Dieppe, P., Macintyre, S., Michie, S., Nazareth, I., & Petticrew, M. (2008). Developing and evaluating complex interventions: the new Medical Research Council guidance. BMJ, 337, a1655. Moore GF, Audrey S, Bond L, Bonell C, Hardeman W, Moore L, O'Cathain A, Tinati T, Wight D, Baird J. Process evaluation of complex interventions: Medical Research Council guidance, BMJ 2015; 350:h1258, doi:10.1136/bmj.h1258.
4	The methods section could be tightened, focusing on your research questions.	See response to comment 2.
5	The setting can be explained more in detail. Could you explain more about the high level of disadvantage? How did you define the level of disadvantage? Also, you should use the complete sentence for the body of the main text.	Thank you for this suggestion. We now use a complete sentence to outline the setting and cross reference to the section in the manuscript that provides detail on how level of disadvantage was determined, to avoid replication of information. This section now reads: "Two Health Board areas in Scotland (Sites A and B) with high levels of disadvantage were the setting for this study (see participant recruitment section for additional details).", p.8.
6	Tables 1 and 3: Did you detect any	We deliberately do not examine statistical

	significant differences between groups (SMS+I vs. SMS only) at baseline, 3-, 6-, and 12-month? The statistical analyses are somewhat brief.	differences in line with CONSORT recommendations for pilot and feasibility trials (Eldridge et al., 2016). However, we did state in the statistical analysis plan that we would report unadjusted mean differences (with 95% confidence intervals, from which statistical significance can be ascertained), but this was pre-specified only for a future full-trial primary outcome. Further reporting of inferential statistics would be inappropriate as this is a pilot study which was not powered for such analyses. Reference. Eldridge, S. M., Chan, C. L., Campbell, M. J., Bond, C. M., Hopewell, S., Thabane, L., & Lancaster, G. A. (2016). CONSORT 2010 statement: extension to randomised pilot and feasibility trials. Pilot and Feasibility Studies, 2(1), 64 We have amended the following sentence in the discussion to make this explicit, and now provide a reference to the above cited CONSORT recommendations: “This feasibility trial was not powered to detect effects on weight loss and, in line with recommendations⁴¹, no p-values are reported. Weight outcomes should be interpreted with caution.”
7	It would be strengthened by organizing the results based on their research objectives/questions.	See response to comment 2.
8	What kinds of limitations of this study did you have? The weaknesses were discussed, but it would have strengthened by adding more aspects of some threats of internal and external validity issues for this study.	In line with comment 1 by review 2 we have added the following limitation in the discussion: “The difference in assessment schedules for intervention and control groups meant that study staff were only partially blind. A full trial should ensure full blinding of outcome assessments”, p.21.
9	Also, it would have enhanced by adding implications for practice and future research.	We state the main implication at the end of the manuscript: “A full trial of the study is warranted”, p.23. Given the feasibility nature of the study we would like to refrain from providing speculation. There are no implications for practice until definitive evidence is available.
10	This manuscript is well-written, and the topic is significant and interesting. I believe the authors can improve the quality of the report.	We thank the reviewer for their helpful comments.

Reviewer: 2

#	Comment	Response
1	Having assessment staff who are not blinded is a significant threat to internal validity. It is clear that blinding was possible but it is not clear why it was not part of the procedures.	We agree with the reviewer. We report on having tested blinding on p.9. Given the feasibility nature of the study and resource implications for instating full blinding of outcome assessors, we opted to test the feasibility of blinding only in this study. We have extended the section on blinding in the limitation section in the discussion on p.20: “Most assessments were not blind to group allocation. The difference in assessment schedules for intervention and control groups meant that study staff were only partially blind. A full trial should ensure full blinding of outcome assessments.”, p.21.
2	The lack of assessment of control participants at 6 months effectively unblinded all staff to the control participants. This is another methodological issue that threatens internal validity.	Thank you for pointing this out. We report on the difference in assessment schedules in relation to blinding on p.9. We have extended the section on blinding in the discussion – see response to comment 1.
3	It is not clear what determined the frequency of text messages.	We have extended the following sentence to address this point on p.10: A narrative text message library consisting of 604 texts was written by a professional scriptwriter/researcher (MG) who designed the overall narrative with enough interlinked stories to engage participants over 12 months.
4	The website seems to be a component of the intervention. It does not appear to be a part of the inclusion criteria and it is not used to assess feasibility.	The webpage was an optional component of the intervention that participants could access if they wanted to. Internet access was not an inclusion criterion to ensure inclusiveness across the socioeconomic spectrum and minimise study complexity. Researchers gave a hard copy of the BDA Weight Loss Food Fact Sheet to all participants which covered weight loss information that was linked to on the Game of Stones website. We provide details on the acceptability of the study webpage on p.19 under the heading “Acceptability of study webpages and pedometer”.
5	Why were control participants interviewed? I realize that detailed information on the interviews is provided in another manuscript, however, if these interviews are relevant for	Guidance on qualitative research in feasibility studies for a randomised controlled trial state that interviews with control group participants are important to assess acceptability and to optimise

	feasibility and acceptability, more information on the interview questions are needed.	trial retention across all groups (O’Cathain et al., 2015). Reference: O’Cathain, A., Hoddinott, P., Lewin, S., Thomas, K. J., Young, B., Adamson, J., ... & Donovan, J. L. (2015). Maximising the impact of qualitative research in feasibility studies for randomised controlled trials: guidance for researchers. Pilot and Feasibility Studies, 1(1), 32. We have added information on the interview questions for the control group participants to the manuscript: “Most control group participants (30/36, 83%) attended the 12 month appointment. Six qualitative interviews were conducted with men in the control group covering topics including acceptability of waiting 12 months for text messages, views about the study and improvement suggestions.”, p.19.
6	There are no predefined standards for feasibility or acceptability. Therefore, it is difficult to determine if the study was feasible or acceptable.	Thank you for this important point. We have now added the pre-specified progression criteria to the manuscript. These were previously referenced in the manuscript as part of the published protocol. The progression criteria are now mentioned as the primary outcome of the trial, p.12. This section now reads: “Outcomes The outcomes for this study related to whether the design of Game of Stones was both acceptable and feasible to deliver as a full scale randomised controlled trial. An independent study steering committee advised whether the following pre-specified progression criteria in the study protocol were met sufficiently to proceed to a full trial.  1. Acceptability of the intervention and the control group (by the majority of the target group); willingness to be randomised. 2. Feasibility of recruiting 105 men in four months. 3. 12-month outcomes on at least 72 % of men randomised per group. 4. Evidence of mean weight loss of at least 3% of baseline weight at 12 months in any intervention group. 5. Commitment by, for example, government or NHS/local authorities to fund the incentive intervention to ensure translation and

		sustainability.” Furthermore, we have added a section to the results outlining that these criteria have been sufficiently met, as judged by the independent study steering committee. “Progression to full trial An independent study steering committee agreed that the Game of Stones study had demonstrated acceptability and feasibility agreeing that overall the pre-specified progression criteria were sufficiently met to support a full three-arm multi-site RCT.”, p.20. In addition, if the editor agrees, we would like to include the table appended to this document as an online supplement to provide further information on how the study findings mapped onto the progression criteria.
7	Lines 18-22 and 28-31 on page 18 appear to be contradictory.	We believe this comment refers to the following lines: “At 12 months, 23/36 participants attended the appointment, with three achieving >10% weight loss securing £200, and a further seven losing between ≥5-10% of weight securing between £100-200.” “No participant secured money at three or six months which was then lost due to weight regain to baseline weight at 12 months.” We rephrased the last sentence to increase clarity: “All participants who completed the study and secured money at three or six months received it at the end of the study. No participant lost previously secured money due to weight regain to baseline weight at 12 months. One participant secured money at an early appointment but withdrew from the study, did not attend at 12 months, and did not receive any money.”, p.19.
8	The authors need to discuss what steps are next given their interpretation of the findings. Where does this study lead them? What are future steps for other researchers in this space? What does this study tell us?	We state that “A full trial of the study is warranted” at the end of the manuscript, p.23. We believe that this is the main future step and would like to refrain from providing speculations on the basis of findings from a feasibility study.

VERSION 2 – REVIEW

REVIEWER	Robert L. Newton, Jr. Pennington Biomedical
REVIEW RETURNED	19-Nov-2019

GENERAL COMMENTS	This is an interesting study, and I think it has more to offer than currently written. My suggestions are designed to be constructive. 1. It is stated that participants received 0-5 text messages per day. What did the range of text messages depend on? Why didn't all participants receive the same number of text messages? 2. The foundation upon which this study rests is there assertion that the study is feasible and acceptable. Therefore, the criteria for feasibility and acceptability should be explicitly clear. I still believe that some of the criteria for acceptability are not clear, and therefore, it is still difficult to determine if they were achieved. a. For example, "Acceptability of the intervention and the control group." It appears that this is operationalized as "Willingness to be randomized." But there are no quantifiable metrics. For example, they randomized over one hundred men, but how many refused? Would the study be deemed acceptable if 50 men agreed? 200? With what proportion of refusals? b. Receiving outcome data on 72% of participants per groups seems arbitrary. What is the rationale for selecting 72%? c. There are no quantifiable acceptability criteria for the main components of the study, the incentives and text messages. The authors report scores of ~3.5 for the text messages and narratives, but the level that defines "acceptability", (e.g. 3.0 on a scale of 1-5; < XX% "stop" messages, etc.), was not provided. Therefore, the acceptability is subjective. d. Relatedly, acceptability of the incentives is not quantifiable. Much of what is listed under "Acceptability" is actually achievability. 3. In the Discussion, the authors describe the incentive strategy utilized in other studies, but they do not provide acceptability or feasibility data. It would be helpful to provide this information in order to better place the current study in context. It will also help support any suggestions for future incentive work. I say this because the data in the current study does not provide strong support for incentives in terms of acceptability and feasibility. 4. The Discussion is still narrow. While I understand that a full trial is the goal of the study, the study has implications for the broader scientific community. What do they learn about this study other than that the authors believe that the study warrants a full trial? How does this study impact the field? What contribution does it make?
---

VERSION 2 – AUTHOR RESPONSE

Reviewer 2

#	Comment	Response
---	---------	----------

1	It is stated that participants received 0-5 text messages per day. What did the range of text messages depend on? Why didn't all participants receive the same number of text messages?	Thank you for this comment. 0-5 is the range of text messages that were sent to participants on a particular day. The number of messages sent on any one day depended on the requirements of the narrative. We have added the sentence: "Texts were sent between 8:00AM and 10:00PM and ranged from 0-5 texts per day depending in the requirements of the narrative approach used. All participants were scheduled to receive the same number of text messages." (p. 10) to the manuscript to further clarify this.
2	The foundation upon which this study rests is the assertion that the study is feasible and acceptable. Therefore, the criteria for feasibility and acceptability should be explicitly clear. I still believe that some of the criteria for acceptability are not clear, and therefore, it is still difficult to determine if they were achieved.	This comment and subsequent comments all refer to how language relating to "acceptability" and "feasibility" are conceptualised and defined. As stated in the manuscript, we used the CONSORT extension for pilot and feasibility studies reporting guidance (Eldridge et al., 2010). This guidance states that "a feasibility study for a future definitive RCT asks whether the future trial can be done, should be done, and, if so, how" (P1) In a separate publication the CONSORT guidance authors discuss how the language for acceptability and feasibility is used differently when reporting pre-trial studies (Eldridge et al., 2016). In line with this guidance, we have outlined the pre-specified criteria for judging acceptability and feasibility of this study, which are included in the manuscript. These criteria were agreed with the funder as part of the peer reviewed grant award process. They were included in the study protocol which was published on the publicly available funders webpage at the start of the study. An independent study steering committee assessed the acceptability and feasibility of this study against these criteria. We will therefore not be able to change these criteria post hoc as part of the revision process of this manuscript. To increase the clarity of the guidance that we followed for this study we have moved the following sentence to the beginning of the methods section and have added the Eldridge et al., (2016b) reference referring the relevant definitions of key terms: "CONSORT guidance for reporting randomised pilot and feasibility studies was followed^{27 28}.", p.7. References. Eldridge SM, Chan CL, Campbell MJ, et al. CONSORT 2010 statement: extension to

		randomised pilot and feasibility trials. BMJ. 2016;355:i5239. Eldridge SM, Lancaster GA, Campbell MJ, et al. Defining Feasibility and Pilot Studies in Preparation for Randomised Controlled Trials: Development of a Conceptual Framework. PLoS One. 2016;11(3):e0150205.
2a	For example, “Acceptability of the intervention and the control group.” It appears that this is operationalized as “Willingness to be randomized.” But there are no quantifiable metrics. For example, they randomized over one hundred men, but how many refused? Would the study be deemed acceptable if 50 men agreed? 200? With what proportion of refusals?	We have further clarified that “Willingness to be randomised was assessed by recording the number of participants refusing randomisation.”, p.13 in the outcome assessment section. In the results, we further clarified that “All 105 men were willing to be randomised and consented to be randomised to the SMS+I (n=36), SMS only (n=33) and control (n=36) groups.” Regarding the request for quantifiable metrics for progression criteria (this comment and comment 2c and 2d) it should be noted that we pre-specified key quantitative acceptability metrics (i.e. weight change, and study retention), with justification. This is a mixed methods study and progression criteria were assessed against qualitative and quantitative information. Assessment of whether other progression criteria were met were informed by using a number of metrics, including quantitative and qualitative information. We did not pre-specify arbitrary criteria for every quantitative outcome that was assessed, and instead employed an independent study steering committee to assess the acceptability and feasibility as a whole, given the obtained findings.
2b	Receiving outcome data on 72% of participants per groups seems arbitrary. What is the rationale for selecting 72%?	Thank you for pointing this out. We have added further rationale for the criterion of 72%, in line with our pre-specified protocol. We have now added “72% of men randomised per group, consistent with a recent UK weight management trial in men³⁴ and systematic reviews of male obesity literature⁴ⁿ, p. 12.
2c	There are no quantifiable acceptability criteria for the main components of the study, the incentives and text messages. The authors report scores of ~3.5 for the text messages and narratives, but the level that defines “acceptability”, (e.g. 3.0 on a scale of 1-5; < XX% “stop” messages, etc.), was not	See response to comment 2a. The reviewer is correct that the decision as to whether a study is acceptable and feasible is a judgement. We delegated this to the independent study steering committee who were appointed by the study funder and who were accountable to the

	provided. Therefore, the acceptability is subjective.	funder.
2d	Relatedly, acceptability of the incentives is not quantifiable. Much of what is listed under "Acceptability" is actually achievability.	See response to comment 2. The current study is an acceptability and feasibility study which is consistent with published recommended terminology for pilot and feasibility studies (Eldridge et al., 2016). Eldridge and colleagues state on P3: "Some might argue that the focus of their study in preparation for a future RCT is acceptability rather than feasibility, and indeed, in other frameworks, such as the RE-AIM framework [53], feasibility and acceptability are seen as two different concepts. However, it is perfectly possible to explore the acceptability of an intervention, of a data collection process or of randomisation in order to determine the feasibility of a putative larger RCT. Thus the use of the term 'feasibility study' for a study in preparation for a future RCT is not incompatible with the exploration of issues other than feasibility within the study itself." The guidance which we used on pilot and feasibility trials does not mention the term "achievability" (Eldridge et al., 2016a; Eldridge et al., 2016b) and neither did our study protocol. Therefore, we do not think that using a new concept of "Achievability" would be helpful to readers. References. Eldridge SM, Chan CL, Campbell MJ, et al. CONSORT 2010 statement: extension to randomised pilot and feasibility trials. BMJ. 2016;355:i5239. Eldridge SM, Lancaster GA, Campbell MJ, et al. Defining Feasibility and Pilot Studies in Preparation for Randomised Controlled Trials: Development of a Conceptual Framework. PLoS One. 2016;11(3):e0150205.
3	In the Discussion, the authors describe the incentive strategy utilized in other studies, but they do not provide acceptability or feasibility data. It would be helpful to provide this information in order to better place the current study in context. It will also help support any suggestions for future incentive work. I say this because the data in the current study	Thank you for this suggestion. The studies that we compare the incentive strategies to are not acceptability and feasibility studies, and we would like to refrain from speculating as to whether these have been acceptable and feasible. Our endowment incentives with SMS for weight loss in men is novel. Therefore, there are no other relevant acceptability and feasibility publications

	does not provide strong support for incentives in terms of acceptability and feasibility.	to discuss.
4	The Discussion is still narrow. While I understand that a full trial is the goal of the study, the study has implications for the broader scientific community. What do they learn about this study other than that the authors believe that the study warrants a full trial? How does this study impact the field? What contribution does it make?	We have revised our conclusion to summarise the key contribution that our study makes in the light of this comment. The conclusion in the manuscript has been changed from: "This three-arm weight management feasibility trial recruited men from across the socioeconomic spectrum with the majority coming from disadvantaged areas, had an acceptable retention rate, and was broadly acceptable to most participants, and feasible to deliver. A full trial of the study is warranted." To: "This study tests a novel combination of narrative SMS with endowment incentives that have not previously been combined in this way to address weight loss and weight loss maintenance in men with obesity. This three-arm feasibility trial recruited men from across the socioeconomic spectrum with the majority coming from disadvantaged areas, had an acceptable retention rate, and was broadly acceptable to most participants, and feasible to deliver. Acceptability and feasibility progression criteria were met. A full trial is warranted to determine effectiveness and cost-effectiveness, with consideration of scalability.", p. 23. We would like to refrain from providing additional speculations of contributions on the basis of a feasibility study.

VERSION 3 - REVIEW

REVIEWER	Robert L. Newton, Jr. Pennington Biomedical Research Center, USA
REVIEW RETURNED	02-Jan-2020

GENERAL COMMENTS	My concerns have been addressed.
----------------------------------